# PARP Inhibitors in Melanoma—An Expanding Therapeutic Option?

**DOI:** 10.3390/cancers13184520

**Published:** 2021-09-08

**Authors:** Wei Yen Chan, Lauren J. Brown, Lee Reid, Anthony M. Joshua

**Affiliations:** 1The Kinghorn Cancer Centre, St Vincent’s Hospital Sydney, Sydney, NSW 2010, Australia; wei.chan@svha.org.au (W.Y.C.); lauren.brown@svha.org.au (L.J.B.); lee.reid@svha.org.au (L.R.); 2Faculty of Medicine, University of New South Wales, Sydney, NSW 2052, Australia; 3Garvan Institute of Medical Research, Sydney, NSW 2010, Australia; 4Melanoma Institute of Australia, Sydney, NSW 2016, Australia

**Keywords:** melanoma, PARP inhibitor, immunotherapy, DNA damage response, homologous recombination, combination therapy

## Abstract

**Simple Summary:**

Melanomas with homologous recombination DNA damage repair pathways represent a subset of melanoma that may benefit from PARP inhibitors and immunotherapy. PARP inhibitors have an established role in treating cancers with underlying *BRCA* mutation through synthetic lethality; however, there is increasing evidence that it can be applied to a larger population including other types of homologous recombination defects. These gene mutations can be found in 20–40% of cutaneous melanoma. To date, PARP inhibitors and immunotherapy have been overlooked in the management of melanoma. This review explores the reasons for combining PARP inhibitors and immunotherapy. There is evidence to suggest that PARP inhibitors can improve the therapeutic effect of immune checkpoint inhibitors. Therefore, this combination approach has the potential to impact future treatment of patients with melanoma, particularly those with homologous recombination DNA damage repair defects.

**Abstract:**

Immunotherapy has transformed the treatment landscape of melanoma; however, despite improvements in patient outcomes, monotherapy can often lead to resistance and tumour escape. Therefore, there is a need for new therapies, combination strategies and biomarker-guided decision making to increase the subset of patients most likely to benefit from treatment. Poly (ADP-ribose) polymerase (PARP) inhibitors act by synthetic lethality to target tumour cells with homologous recombination deficiencies such as *BRCA* mutations. However, the application of PARP inhibitors could be extended to a broad range of *BRCA*-negative cancers with high rates of DNA damage repair pathway mutations, such as melanoma. Additionally, PARP inhibition has the potential to augment the therapeutic effect of immunotherapy through multi-faceted immune-priming capabilities. In this review, we detail the immunological role of PARP and rationale for combining PARP and immune checkpoint inhibitors, with a particular focus on a subset of melanoma with homologous recombination defects that may benefit most from this targeted approach. We summarise the biology supporting this combined regimen and discuss preclinical results as well as ongoing clinical trials in melanoma which may impact future treatment.

## 1. Introductions

Treatment of metastatic melanoma has been revolutionised over the last decade. The use of targeted therapies and checkpoint inhibitors have significantly improved long-term outcomes. Although progress has been made in targeting melanoma with pre-requisite genotypes using small molecule inhibitors such as BRAF/MEK inhibitors, most relapse after 6 to 9 months as the majority will develop drug resistance [1]. Furthermore, 40–60% of patients with melanoma have de-novo or acquired resistance to immunotherapy leading to disease progression [2,3]. Hence, the need for new therapies is an evolving clinical challenge. Multigene panel sequencing is routine in many solid tumours but has been relatively limited in melanoma due to the immediate clinical impact of *BRAF* assessment in directing clinical care. However, larger panel testing can facilitate deeper understanding of solid organ tumours, including melanoma, and expand pathways for targeted therapies. 

Melanomas with homologous recombination DNA damage repair (HR-DDR) represent a subset of melanoma that may benefit from these targeted treatment options including the addition of poly (ADP-ribose) polymerase (PARP) inhibitors in combination with immunotherapy. The use of DNA damage repair (DDR) agents, such as PARP inhibitors, appear to activate the immunosuppressive pathways of homologous recombination (HR) tumours, thus providing a targetable immunologic vulnerability which forms the basis of combination therapy with PARP inhibitors and immune checkpoint inhibitors (ICI). In this review, we summarise homologous recombination deficiencies in melanoma and the evolving therapeutic options for these patients, including harnessing potential synergies between PARP inhibitors and immunotherapy. MEDLINE PubMed and EMBASE databases were searched for relevant articles including the keywords melanoma, homologous recombination deficiency, DNA damage repair, PARP inhibitor, immunotherapy and combination therapy.

## 2. Subgroups of Melanoma

Traditionally, melanoma has been classified based on primary tissue type. The major subtypes include cutaneous melanoma, acral melanoma, mucosal melanoma and uveal melanoma [4]. Of these subtypes, mucosal melanoma appears to be the most biologically aggressive with the worst prognosis [5]. Melanoma can be further classified into four genetic subtypes, based on the absence or presence of driver mutations [6]. These genomic subtypes include *BRAF*-mutant melanoma (50%), *NRAS, KRAS* and *HRAS*-mutant melanoma (25%), *NF1*-mutant melanomas (15%) and triple wild-type melanomas (10%) [7]. Other genetic abnormalities commonly found in melanoma include activating *TERT* promoter variations found in 30–80% of melanomas and tumour suppressors such as *CDKN2A, PTEN, TP53* and *ARID2* [7]. 

Melanoma from sites of sun damaged skin have been demonstrated to have a higher mutational burden than tumours from non-chronically sun damaged sites [8]. The most frequent somatic mutations in sun-exposed melanomas involve genes that control cellular processes such as proliferation (*BRAF, NRAS, NF1*), metabolism and growth (*PTEN, KIT*), resistance to apoptosis (*TP53, CDKN2A*) and replicative lifespan (*TERT*) [9]. These genomic alterations lead to activation of the two main signalling cascades in melanoma: the RAS/RAF/MEK/ERK (aka MAPK) pathway and the PI3K/AKT pathway; both drivers of cell proliferation and homeostasis [9]. Up to 98% of these melanomas have been found to exhibit abnormalities in the MAPK pathway [8]. A significant portion (51%) of mucosal and acral melanomas were triple wild-type, meaning they lacked *BRAF, NRAS* or *NF1* mutations, whereas only 11% of cutaneous melanomas were found to be triple wild-type [8,10]. *BRAF, NRAS* and *TERT* mutations can be found in both benign lesions and malignant melanoma; however, *CDKN2A, TP53* and *PTEN* were exclusively found in malignant lesions [8]. 

Sequencing of other melanocytic neoplasms have also been critical in identifying other gene fusions with potential targets [11]. In Spitzoid neoplasms, *ALK* fusions were found in 10–20% and *NTRK1* fusions in 20% [11]. In Spitz melanomas, *ROS1* fusions were observed in 9% of patients, whilst some observed a *MET* fusion [11]. These targets have been established in other solid organ tumours and could serve as potential therapeutic agents for specific subtypes of melanocytic neoplasms. 

## 3. Description of DNA Repair Damage Process and PARP Synthetic Lethality

The DNA damage response pathway refers to a class of proteins that repair DNA damage such as translocation, deletions, double strand breaks, chromosomal fragmentation and various alterations [12]. DNA damage or replication stress will mobilise transducers, effectors or signal sensors, leading to either tolerance or correction of damaged DNA [13]. There are three main repair pathways for DNA single strand breaks: (1) mismatch-mediated repair (MMR); (2) base excision repair; (3) nucleotide excision repair, and two main repair pathways for double-strand DNA breaks: (1) homologous recombination and (2) more error prone non-homologous end joining (NHEJ) [14]. 

Homologous recombination is composed of multiple interrelated pathways that function to repair DNA double-stranded breaks and provide support for DNA replication, contributing to tolerance of DNA damage [15,16]. Therefore, inability to repair complex DNA damage, such as mutations in *BRCA1* and *BRCA2* tumour suppressor genes, can lead to genomic instability and predisposition to cancers [16]. 

PARP facilitates DNA repair by binding to sites of DNA damage, catalysing poly ADP-ribose chains and subsequently recruiting effector proteins, modifying damaged chromatin and acting as an energy sink [17]. The inhibition of this process is thought to be a canonical mechanism of action of PARP inhibitors. The second mechanism of action of PARP inhibitors is PARP trapping where activated PARP molecules are trapped onto damaged DNA, thereby blocking DNA repair proteins from DNA replication and leading to accumulation of single-stranded DNA breaks, induction of double strand breaks and cell death [18,19]. More recently, a third and possibly unifying mechanism has been proposed for PARP sensitivity that relates to replication gap formation, a phenotype that is rescued by loss of p53-binding protein 1 (53BP1) [20]. While cells that have intact HR are capable of repairing the double-stranded breaks that are created by PARP inhibitors, cells with mutations in HR, such as defective *BRCA1* or *BRCA2* or other DDR genes, are unable to effectively perform HR, thereby causing cell death with synthetic lethality [21,22] (Figure 1). 

Although *BRCA1* and *BRCA2* mutant tumours are the best-known associations of HR deficiency and intrinsically sensitive to PARP inhibitors, there are a wide range of other non-*BRCA* DNA repair genes associated with HR deficiency, including but not limited to *ARID1A, ATM, PALB2, CHEK2* and *FANCA* [23]. Thus, PARP inhibitors may have utility beyond the small proportion (5–10%) of patients carrying *BRCA* mutations [24]. To date, PARP inhibitors have established a role in treatment of patients with HR deficiency and/or *BRCA* mutant tumours such as prostate, breast and ovarian cancers [23].

## 4. Homologous Recombination in Melanoma 

The prevalence of HR-DDR amongst tumour lineages is poorly characterised even though various solid tumours have been treated with PARP inhibitors [25]. There is a need to understand the clinical characteristics of melanoma patients with HR-DDR gene mutations and their impact on treatment decisions [23].

### Frequency of HR Mutations in Melanoma

Homologous recombination deficiency can be found in almost all types of cancers. There have been varying reports of frequency in cutaneous melanoma, such as Kim et al. which reported that 21.4% harboured a HR-DDR gene pathway mutation [23]. In this cohort, the presence of an HR-DDR pathway gene mutation was associated with a significantly higher proportion of thinner primary tumours, head and neck primary tumours, higher tumour mutational burden (TMB) and concurrent *NF1* mutation [23].

On the other hand, the Foundation Medicine Cohort reported a prevalence of 33.5% and the cBioportal cohort reported 41% with at least one HR-DDR pathway gene mutation [23]. A pan-cancer analysis of 52,426 patients across multiple solid tumours reported an 18.1% prevalence of HR-DDR mutations in melanoma [25]. Variances in frequency are attributed to differences in testing platforms, the gene sets used to define HR-DDR, and the focus of cutaneous melanoma in each study. Nevertheless, all four analyses indicate that HR-DDR mutations are common events in melanoma [25]. Each dataset also reported varying frequencies of the most commonly mutated HR-DDR genes in melanoma, as summarised in Table 1 [23]. The prevalence of HR-DDR mutations in melanoma are compared to other tumour types based on molecular profiles generated through next-generation sequencing with NGS600 in Heeke et al. (Figure 2) [25].

In fact, a number of HR-DDR genes, such as *BRCA1/2, FANCA* and *BAP1*, are associated with genetic syndromes that increase predisposition to cancers such as melanoma [26]. For instance, *BAP1* (*BRCA1*-associated protein 1) is a tumour suppressor that regulates cell cycles, cell differentiation and DNA damage response pathways [27]. Germline and somatic *BAP1* mutations confer increased risk of developing cutaneous melanoma, uveal melanoma, epithelioid atypical Spitz tumours, clear cell renal cell carcinomas and mesothelioma [26,27]. High penetrance genes can also increase the risk of melanoma, as best seen in familial melanoma which represents 5–10% of all cutaneous melanomas [28,29]. Approximately 40% of familial cases can be attributed to high penetrance melanoma susceptibility genes such as *CDKN2A, CDK4* and *BAP1* [28].

## 5. Evidence in Xenograft Models and Cell Lines of PARP Inhibition in Melanoma

There have been preliminary studies in the use of PARP inhibition in melanoma cell lines with no HR deficiency, cell lines with HR deficiency in vitro and in patient-derived xenograft models. In vitro studies of niraparib treatment on melanoma cell lines, MM425X, (harbouring mutations in *BRCA1* and *ARID1B*), MM390 (harbouring mutation in *CHD2*) and MM507X cells (no detectable HR mutations) at concentrations between 1 um and 60 um in cultures showed decreased survival and induction of apoptosis in both of the HR-DDR positive samples [23]. However, niraparib had minimal effect on survival in the MM507X cell lines. 

Patient-derived xenograft models were also utilised to assess the use of PARP inhibition in murine models. The same cell lines were evaluated in NSGTM mice. A dose finding study for niraparib and Olaparib showed a dose of 25 mg/kg for niraparib and 50 mg/kg for Olaparib resulted in an anti-tumour effect for MM425X [23]. RNA sequencing on the MM425 *BRCA1* sample from three mice treated with intraperitoneal niraparib showed changes in cell cycle, movement, integrin signalling, collage and matrix remodelling and triglyceride and fatty acid metabolism [23].

Mice treated with PARP inhibitors were found to have a decreased burden of metastatic disease [30]. Fewer endothelial cell markers were measured in the metastatic foci, suggesting that PARP inhibition also plays a role in decreasing tumour angiogenesis. Both of these in vitro studies and murine models have demonstrated strong pre-clinical evidence for the role of PARP inhibitors in the treatment of melanoma, in particular HR-DDR positive disease.

## 6. PARP Inhibitors in Melanoma

The selective efficacy of PARP inhibitor monotherapy has been established in tumours harbouring defects in *BRCA* or other genes required for repair by homologous recombination, such as ovarian, prostate, pancreatic and breast cancer; however, there has been limited data to date for melanoma [20]. Given the preclinical data and growing evidence of the utility of PARP inhibitors in xenograft models, there is a strong rationale to evaluate the clinical efficacy of PARP inhibitors in patients with advanced melanoma with homologous mutations, alterations or deficiency. Over-expression of DNA repair genes in melanoma has been associated with increased rates of relapse and a lower likelihood of response to chemotherapy [31]. Therefore, PARP has been investigated in combination with cytotoxic chemotherapy, particularly temozolomide, in attempts to overcome resistance in alkylating agents.

A phase II, double-blind trial of patients with unresectable stage III or IV metastatic melanoma were randomised 1:1:1 to temozolomide plus veliparib 20 mg or 40 mg, or placebo [32]. This trial of 346 patients demonstrated a trend towards improvement in progression-free survival (PFS) (median PFS 3.7 vs. 3.6 vs. 2 months), which was not statistically significant, and no difference in overall survival [32]. The authors noted the need for predictive biomarkers to be explored in future studies to identify subsets of patients that may benefit from PARP inhibitors. Another single-arm phase II study of rucaparib and temozolomide in advanced metastatic melanoma also showed improvement in progression-free survival but did not reach statistical significance [33]. The response rate was 17.4%, the median time to progression 3.5 months, and the median overall survival 9.9 months [33]. However, myelosuppression was the dose-limiting toxicity of this combination, although no toxicity was attributable to the PARP inhibitor alone [33]. These two phase II studies showed enhanced bone marrow suppression, requiring an 80% dose reduction to safely deliver the combination [33].

With a more targeted approach in HR deficient melanomas, there may be more promise given a greater therapeutic ratio. Recently, a case report described the use of single-agent Olaparib in a patient with somatic *PALB2* mutation with metastatic melanoma [34]. The patient, having previously progressed on combination immunotherapy (Ipilimumab and Nivolumab), demonstrated a partial response to the PARP-inhibitor Olaparib [34]. The treatment response was ongoing at 6 months and highlights the importance of testing for homologous recombination defects in melanoma patients [34].

## 7. Immunologic Role of PARP 

Monotherapy with PARP inhibitors has shown clinical activity as a maintenance strategy in ovarian, breast and pancreatic cancer harbouring germline loss-of-function *BRCA* mutations [35,36]. Immune checkpoint inhibitors (ICI) have also demonstrated an ability to induce durable and high response rates in a range of cancer subtypes including melanoma, renal cell carcinoma, non-small cell lung cancer and urothelial carcinoma [37]. However, despite significant improvement in patient outcomes, most patients receiving either PARP inhibitors or immunotherapy alone do not derive benefit [35].

Melanoma is one of the most sensitive malignancies to immune system modulation [38]. This may be explained by several factors including high tumour mutational load from ultraviolet light exposure, mimicry of melanocyte proteins with pathogen-associated antigens and recognition of melanoma antigens by tumour infiltrating lymphocytes (TILs) [38,39]. Most importantly, the high immunogenicity of melanoma lends to the effectiveness of immunotherapy in its treatment strategy [38,39].

Patients with metastatic melanoma treated with ipilimumab and nivolumab still have a high rate of progressive disease (64% at 5 years); hence, there is an unmet need for novel therapeutics in this patient population [40,41]. Only a minority (20%) of patients with melanoma derive long-term response, while the remaining develop primary or secondary resistance [39]. Therefore, better understanding of the determinants of response to ICIs and identification of combinations that would increase the proportion of patients benefiting from these therapies are crucial [42]. Herein, we will explore some of the associations of PARP inhibition with the immune response as well as the potential for synergy with existing checkpoint inhibitors. 

### 7.1. Rationale for Combining PARP and Immune Checkpoint Inhibitors

There is increasing evidence that shows the addition of PARP inhibitors can increase the immune response of ICI therapy. Defects in DNA damage response results in genomic instability and increased tumour mutational burden, which are key determinants in cancer immunogenicity [42]. Beyond maintaining genomic stability, PARP plays a significant role in both innate and adaptive immune responses, and influences anti-tumour immunity via T cells [43]. The unrepaired DNA promotes immune priming through a range of molecular mechanisms and leads to adaptative upregulation of PD-L1 expression, alteration of inflammatory tumour microenvironment and increased TH1 immune response [14]. This multifaceted immunological effect of PARP inhibitors may be favourable for boosting an antitumor immune response and enhancing the efficacy of ICIs [35]. The rationale for PARP inhibitors in combination with ICIs mainly involves four aspects as discussed below: (1) tumour mutation burden and enhanced neoantigen production; (2) upregulation of PD-L1 and cyclic GMP-AMP synthase-stimulator of interferon (cGAS-STING) pathway (3) reprograming of immune cells involved in tumour microenvironment (TME); (4) increasing tumour-infiltrating lymphocytes [43]. 

#### 7.1.1. Tumour Mutation Burden and Neoantigen 

The relationship between TMB and efficacy of ICI has been established in many prior studies [44,45,46,47]. Highly mutated tumours often contain one or more mutations in key components of DDR pathways, such as *BRCA1/2* for HR, *MSH2* for MMR and *POLE* for DNA replication deficiency [17,35,48]. Patients with these innate deficiencies tend to achieve higher response rates and more durable benefit from ICIs compared to patients without [17]. This suggests that loss of normal DNA repair, such as “BRCAness” or DDR phenotype, affects the therapeutic response to immunotherapy by contributing to increased tumour mutational burden and neoantigen load [35,49].

TMB is often regarded as a surrogate for neoantigen burden [43,50,51,52]. Neoantigens are mutated proteins specifically expressed by tumour tissue and not by normal tissue, making it highly immunogenic, i.e., a critical target for tumour immunotherapy [53,54,55]. By affecting the DDR or HR pathways in tumour cells, PARP inhibition can trigger catastrophic, irreparable DNA damage and tumour cell death which releases neoantigens and increases TMB and immunogenicity [43,56]. Therefore, PARP inhibitors may facilitate a more profound anti-tumour immune response and drive a response to ICI, theoretically extending the population of patients who may respond [43]. 

#### 7.1.2. Upregulation of PD-L1 

The expression of the inhibitory ligand PD-L1 on the surface of tumour cells is not only a vital mechanism for cancer immune invasion but is also an important biomarker in predicting the efficacy of response to ICI [57,58,59]. Increasingly, studies have identified mechanisms through which PARP inhibitors can increase the expression of PD-L1. Upregulation of PD-L1 is primarily driven by tumour-associated inflammation via the cGAS-STING pathway, reflecting the status of TME [17,60]. PARP inhibition increases double-stranded DNA breaks which causes cGAS binding, leading STING activation and generation of a type I interferon (IFN) response [35,43,61]. Type I IFN promotes activation of anti-tumour responses, T cell recruitment and increased PD-L1 expression [35,43]. 

Defects in *BRCA1/2* also correlate to higher levels of PD-L1 expression [35]. In addition, preclinical studies have shown that PARP inhibition can upregulate PD-L1 expression by inactivating GSK-3β, thereby suppressing T-cell activation and increasing tumour cell killing [43,62]. Furthermore, PARP-inhibitor-induced double-stranded breaks can increase PD-L1 through the ATM-ATR-CHK1 pathway independent of the IFN pathway [43,63]. Thus, PD-L1 upregulation by PARP inhibition can theoretically increase sensitivity to ICI and potentially lead to greater anti-tumour activity when combined compared to either drug on its own. 

#### 7.1.3. Reprogramming of Tumour Immune Microenvironment by PARP Inhibitors

Interaction between DDR and the immune response is the foundation for combining PARP inhibitors and ICI [17]. Apart from altering the intrinsic immunogenicity of tumour cells through intracellular pathways and surface phenotypes, DNA damage and DDR deficiency can modify the extrinsic immunogenicity of tumours at a microenvironment level [35]. At baseline, TME causes low-level DNA damage, thus inducing chronic inflammation which promotes the development of cancers [17]. PARP inhibitors have the potential to shift low-level, chronic DNA damage to a more significant Th1 immune response by causing further DNA damage, leading to subsequent acute inflammation, and stimulating production of type I IFN via the cGAS-STING pathway [35,47,64]. Activation of the cGAS-STING pathway remodels the immune response to create a more susceptible TME which boosts immune priming and induces extrinsic tumour suppression [17]. PARP inhibition also influences the TME through regulation of NK cells, production of chemokines, angiogenesis and oxidative stress [43]. Overall, this leads to increased inflammation and T-cell infiltration which enhances the tumour response to the immune checkpoint blockade [14].

Nevertheless, treatment with a single-agent PARP inhibitor is insufficient to exert durable therapeutic effects due to the conflicting impact on tumour microenvironments [35,64]. Although PARP inhibitors produce inflammatory signals which trigger appropriate anti-tumour responses, it can also stimulate myeloid cell recruitment which suppresses immune cells and favours tumour growth [35]. Myeloid cells activate the immune checkpoint PD-1/PD-L1 axis required for an immunosuppressive TME, which counterbalances the therapeutic efficacy of PARP inhibitors alone [64]. This challenge can be addressed by synergising with ICIs to inhibit immunosuppressive myeloid cells [64]. Hence, it is possible that combining a PARP inhibitor with immunotherapy will overcome resistance mechanisms and result in better outcomes.

#### 7.1.4. Increasing Tumour Infiltrating Lymphocytes

A high number of TILs is typically considered indicative of immunogenicity [65]. In pre-clinical *BRCA*-deficient models, PARP inhibition can increase the infiltration of helper (CD4+) T cells and cytotoxic (CD8+) T cells by activating the STING pathway [43]. One preclinical tumour model investigating niraparib with anti-PD1 checkpoint inhibitors showed increased interferon pathways and enhanced infiltration of CD8+ cells and CD4+ cells in tumour cells, resulting in synergistic anti-tumour activity [66]. Studies of *BRCA1/2*-mutated tumours in breast and ovarian cancer have shown higher frequency of TILs compared to HR-proficient tumours [65]. Hence, combined PARP inhibitor and ICI therapy may prolong responses for HR-deficient tumours. 

## 8. PARP Inhibitors and Immunotherapy in Clinical Use

### 8.1. Studies of Combination of PARP Inhibitor and Immunotherapy in Any Solid Tumours

Early development of PARP inhibitors focused on their use in combination with cytotoxic chemotherapy and radio-sensitizing drugs, but this was abandoned quickly due to excessive toxicity [35]. Given the potential synergy between PARP inhibitors and ICI, multiple studies have been developed to explore the clinical applications and efficacy of this combination therapy in tumours harbouring *BRCA1/2* or other DDR gene mutations [17]. 

Of note, there are 35 ongoing studies ranging from phase I to III, combining PARP inhibitors and immunotherapy in solid tumours (predominantly breast, ovarian, gastric, lung, bladder, colorectal, head and neck, prostate, and biliary tract cancers). However, there are only four all-comer studies potentially enrolling melanoma patients (Table 2). Unfortunately, a phase II study (NCT03637491) evaluating talazoparib, avelumab and binimetinib in metastatic RAS-mutant solid tumours was ceased after two years as data showed limited anti-tumour activity and it was not feasible to reach the target study drug dose levels.

To date, only three combinations of PARP inhibitor and immunotherapy have published data, i.e., Olaparib/durvalumab, niraparib/pembrolizumab, and BGB-A317/BGB-290 [14] (Table 3). Exploratory analysis of biomarker subpopulations in ovarian cancer indicated that combination treatment of niraparib and pembrolizumab resulted in antitumour activity and consistent objective response rate (ORR) across the study population regardless of tumour *BRCA* or HRD biomarker status [67]. However, in the TNBC cohort, patients with the *BRCA* mutation benefitted more compared to those without, with an ORR of 47% vs. 11% and a disease control rate (DCR) of 80% versus 33% [68]. In metastatic castrate-resistant prostate cancer, patients with DDR mutations had greater benefit with a combination of Olaparib and durvalumab than those without (median PFS 16.1 months vs. 4.8 months) [69]. Conversely, in monotherapy studies, pembrolizumab showed a 6% PSA response rate and Olaparib alone led to a 22% PSA response rate [70,71]. Median PFS was 9.8 months in DDR-deficient patients and 2.1 months in DDR-proficient patients [70,71].

Both combinations of Olaparib/durvalumab and niraparib/pembrolizumab were well tolerated, with toxicities in line with those observed for the relevant agents in monotherapy settings. In contrast, the BGB-A317/BGB-290 combination demonstrated higher rates of hepatic toxicity, suggesting that tolerability of PARP inhibitor and anti-PD-1/PD-L1 combinations may vary depending on the agents utilized. All three combinations showed evidence of antitumor activity in a range of settings [14].

### 8.2. Studies of Combination of PARP Inhibitor and Immunotherapy in Melanoma 

From a clinical perspective, in patients with melanoma refractory to anti-PD1 therapy, further targeted therapies such as PARP inhibitors are needed. It is postulated that PARP inhibitors together with ICI have a synergistic immunomodulatory and anti-tumour effect. Given we know the efficacy of immunotherapy agents in melanoma, using PARP inhibitors in patients who are refractory to ICI’s may assist in harnessing a therapeutic and immunogenic response resulting in clinical efficacy when combined. 

A recent case report detailed treatment of a patient with metastatic melanoma on combination nivolumab (480 mg intravenously, monthly) and Olaparib (300 mg orally, twice daily) after relapse on maintenance nivolumab, seven months post induction of ipilimumab and nivolumab [76]. The patient had homologous recombination deficiency and several mutations in the DDR pathway (germline *CHEK2* mutation and somatic mutations in *BRCA2, ATRX, TP53, NF1*), and achieved a complete radiologic response on a PET of a metastatic liver lesion two months after commencing combination therapy, with no side effects [76]. Several clinical trials are ongoing to examine the synergistic effects of PARP inhibition and the immune checkpoint blockade in melanoma (Table 4).

## 9. Future Perspectives and Conclusions

The use of immunotherapy in melanoma has transformed its therapeutic landscape with significant improvement in patient outcomes; however, de-novo or development of resistance to immunotherapy still leads to progressive or recurrent disease in many. Therefore, it is paramount to expand therapeutic options for patients with melanoma. PARP inhibitors have demonstrated a clear role in treating tumours with underlying *BRCA* mutations through synthetic lethality, but there is increasing evidence that the application of PARP can be extended beyond *BRCA*-mutant cancers and towards a larger population of patients with the use of homologous recombination repair deficiency as a novel biomarker. PARP inhibitors have been explored as a monotherapy or as a combination approach in various solid tumours with promising results; however, it has been largely overlooked in the management of melanoma to date. 

Cell line and preclinical data indicate synergistic effects of a combination of PARP and immune checkpoint inhibitor therapy. PARP inhibition induces double-strand DNA breaks via the cGAS-STING pathway, leading to genomic instability, increased tumour burden and thus, immunogenicity. Unrepaired DNA also promotes inflammation, immune priming and upregulation of PD-L1 expression. The multiple links between PARP and the tumour–immune response suggest PARP inhibitors are a potential sensitizer for immune checkpoint inhibitors. Therefore, combination therapy has the potential to improve patient responses in melanoma patients with (20–40%) and without HR-DDR defects. At time of this review, there are two phase II clinical trials investigating this combination. Further research is needed to determine the efficacy and safety of this combination, and the role of HR-DDR mutations in identifying the patients who may benefit the most from this combination. 

## Figures and Tables

**Figure 1 cancers-13-04520-f001:**
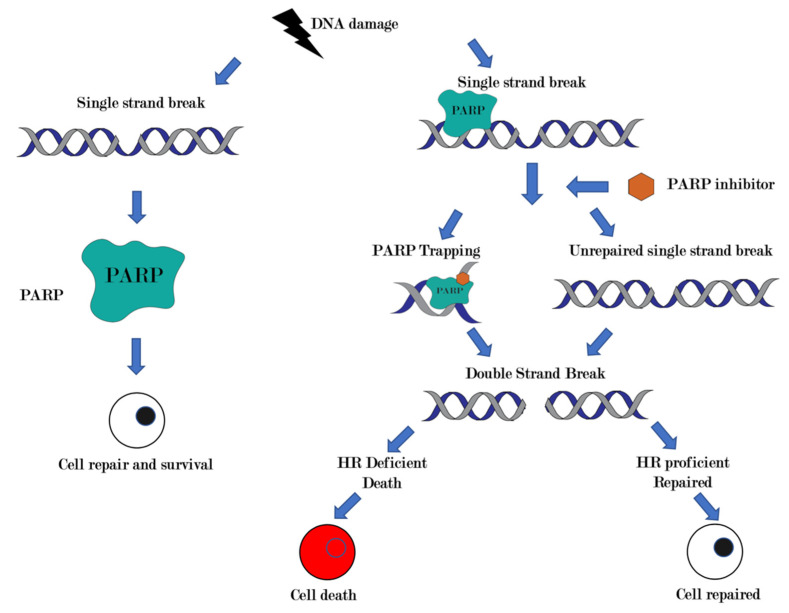
Canonical mechanism of action of PARP inhibitors and synthetic lethality. Poly (ADP-ribose) polymerases are proteins that play a role in DNA damage repair. PARP inhibitors impair base excision repair, which prevents repair of single-strand breaks (SSB). Accumulation of SSB causes replication fork stalling and generates lethal double-stranded breaks (DSB). In homologous recombination (HR) deficient cells, such as *BRCA* mutated cells, double-stranded breaks cannot be repaired efficiently, thus leading to cell death.

**Figure 2 cancers-13-04520-f002:**
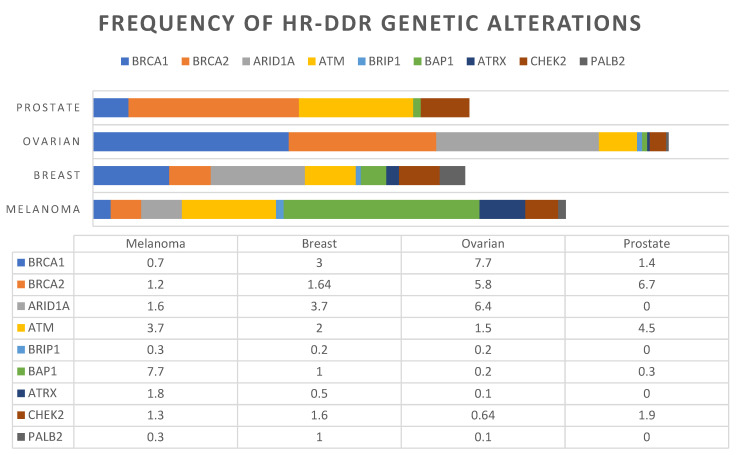
Prevalence of homologous recombination DNA damage repair (HR-DDR) gene mutations according to tumour lineage, in particular prostate, ovarian, breast cancer and melanoma. A total of 17,566 tumours underwent next generation testing with NGS600 to evaluate HR genes based on Heeke et al. [25].

**Table 1 cancers-13-04520-t001:** Frequency of most commonly altered HR-DDR Gene Mutations in Melanoma.

Gene	Function/Pathway	Foundation Medicine (*N* = 1986), %	cBioportal Cohort (*N* = 1088), %	CPMCRI Cohort (*N* = 84), %	Heeke et al. NGS600 (*N* = 17,566), %
*BRCA1*	BRCA	1.3	5	3.6	0.75
*BRCA2*	BRCA/Fanconi	2.3	8	0	1.2
*ARID1A*	Chromatin remodelling	5.0	7	3.6	1.6
*ARID1B*	Chromatin remodelling	0.1	6	1.2	0
*ATM*	DSB repair	4.0	7	2.4	3.7
*ATR*	DSB repair	1.6	7	2.4	0
*FANCA*	Fanconi	1.0	4	2.4	0
*FANCD2*	Fanconi	0.2	5	1.2	0
*ATRX*	Chromatin remodelling	2.8	7	1.2	1.8
*BRIP1*	BRCA/Fanconi	1.1	4	1.2	0.3
*BAP1*	Tumour suppressor	3.1	2.3	1.2	7.7
*CHEK2*	DSB repair	0.7	2.5	1.2	1.3
*BARD1*	BRCA	0.2	1.7	1.2	0
*PALB2*	BRCA/Fanconi	0.5	4	1.2	0.3
*RAD50*	DSB repair	0.9	2.4	0	0
Total		33.5	41	21.4	18.1

**Table 2 cancers-13-04520-t002:** “All-comer” studies combining PARP inhibitors and immunotherapy.

Combination	ClinicalTrials.Gov Identifier	Phase	Cancer	References	Status
Olaparib + Durvalumab	NCT03772561	I	Advanced solid tumours	N/A	Recruiting
Niraparib + TSR-042	NCT03307785	I/II	Advanced solid tumours	N/A	Ongoing
Talazoparib + Avelumab	NCT03565991	II	Tissue agnostic study in BRCA/ATM mutant solid tumours	N/A	Ongoing
Talazoparib + Avelumab	NCT03637491	II	Triplet combination with binimetinib in RAS-mutant solid tumours	N/A	Terminated

**Table 3 cancers-13-04520-t003:** Combination studies with available data.

Combination	ClinicalTrials.Gov identifier	Phase	Cancer	References	Status
Olaparib + Durvalumab	NCT02734004 (MEDIOLA)	I/II	Basket study in gBRCA-mutant ovarian, HER2- breast cancer, relapsed platinum-sensitive SCLC and gastric cancer	[72,73]	Ongoing
Olaparib + Durvalumab	NCT02484404	I/II	Basket study in previously treated ovarian, gBRCA-mutant TNBC, lung, prostate and MSI-S colon	[69,74]	Recruiting
Niraparib + Pembrolizumab	NCT02657889 (TOPACIO)	I/II	HER2- TNBC and ovarian cancer	[67,68]	Ongoing
BGB-A317 + BGB-290	NCT02660034	I	Basket study ovarian, TNBC, mCRPC, bladder, SCLC, HER2- gastric and pancreatic cancer	[75]	Completed

**Table 4 cancers-13-04520-t004:** Ongoing Trials of PARP inhibitors and Immunotherapy in Melanoma.

PARP Inhibitor	Immunotherapy	ClinicalTrials.Gov Identifier	Phase	Eligibility	Ref
Niraparib	Nil	NCT03925350	II	Metastatic melanoma progressed on standard therapied with genetic homologous recombination mutation or alteration	N/A
Olaparib	Pembrolizumab	NCT04633902	II	Advanced melanoma with genetic homologous recombination mutation or alteration resistant to anti-PD-1 therapy	N/A
Talazoparib	Nivolumab	NCT04187833	II	Unresectable or metastatic melanoma patients with mutations in BRCA or BRCAness	[77]

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
