# Peer review of "PARP Inhibitors in Melanoma—An Expanding Therapeutic Option?"

_cancers, 2021, doi:10.3390/cancers13184520_

Round 1
Reviewer 1 Report
An interesting narrative review about Poly (ADP-ribose) polymerase inhibitors in the management of melanoma.
I found the paper very well structured, concise, and eligible for publication after minor revisions. I have some queries:
Although the article is a narrative review, a materials and methods sections stating what databases (Pubmed, Google Scholar, Scopus/Embase, etc...) and what keywords were used in order to select the studies included in the paper would be in my opinion a great addition.
Page 2 line 50 you should add: "mucosal melanoma seems to have the worst prognosis among these subtypes" and cite an article such as doi: 10.3390/medicina57040359.
Thank You
Reviewer 2 Report
The authors provided with their work an extensive overview what is known about the clinical benefit using PARP inhibitors in melanoma. Anyhow some points should be adressed before the work is ready for publication.
1. Introduction:
First paragraph: A sentence on resistance to small molecule inhibitors including matching references should be added.
Second paragraph: I think PARP inhibitors alone are not a therapy option for melanoma.
3. Description of DNA repair damage process and PARP synthetic lethality
Third paragraph: The trapping of the PARP molecules not only leads to unrepaired single strand breaks, but also to the induction of double strand breaks!!
Reference 18 is not talking abaout p53, it is 53BP1.
Figure 1 should be improved by e.g. adding the structure of a PARP molecule. And not only accumulation of SSBs but also PARP that can not dissociate from DNA causes replication fork stalling and gen-107 erates lethal double stranded breaks (DSB).
4. Homologous Recombination in Melanoma
First paragraph: add a sentence to homologous recombination also in this first paragraph, otherwise the headline is missleading.
5. Evidence in Xenograft Models and Cell Lines of PARP inhibition in melanoma
First paragraph: "(harbouring a mutation in CHD2 mutation)" delete the second time mutation
7.1.3 Reprogramming of tumour immune microenvironment by PARP inhibitors
Specify the resistence mechanisms which can be overcome mentioned in the last sentence.
7.1.4 Increasing tumour infiltrating lymphocytes
T-cells in the third line is missing an "s".
Table 2: A column on the current status of the study should be added (ongoing, recruiting, stopped). If the study has already been stopped, a brief discussion of why should be included.
Table 2 and 3: If there is a column "Combination" this should be filled out for every study.
